# Composition and Bioactivity of Chlorogenic Acids in Vegetable and Conventional Sweet Potato Vine Tips

**DOI:** 10.3390/foods12213910

**Published:** 2023-10-25

**Authors:** Fantong Meng, Wantong Du, Yaxing Zhu, Ximeng Du, Chengchuang Song, Xi Chen, Xingtang Fang, Qinghe Cao, Daifu Ma, Yanhong Wang, Chunlei Zhang

**Affiliations:** 1Jiangsu Key Laboratory of Phylogenomics and Comparative Genomics, Institute of Cellular and Molecular Biology, College of Life Science, Jiangsu Normal University, Xuzhou 221116, China; mengfantong863329@163.com (F.M.); 18052186060@163.com (W.D.); zyx15006216001@163.com (Y.Z.); duximmm@163.com (X.D.); chengchuangsong@163.com (C.S.); cxvirus@126.com (X.C.); xtfang11@126.com (X.F.); caoqinghe@jaas.ac.cn (Q.C.); daifuma@163.com (D.M.); 7096871@163.com (Y.W.); 2Sweetpotato Research Institute, Chinese Academy of Agricultural Sciences, Xuzhou 221004, China

**Keywords:** chlorogenic acids (CGA) separation and purification, anti-breast cancer, sweet potato, LC-MS, principal component analyses (PCA), linear discriminant analysis (LDA)

## Abstract

Sweet potato vine tips are abundant in chlorogenic acid (CGA). In this study, CGA was extracted from vegetable and conventional sweet potato vine tips using ethanol, followed by subsequent purification of the extract through a series of sequential steps. Over 4 g of the purified product was obtained from 100 g of sweet potato vine tip powder, producing more than 85% of purified CGA. The LC-MS analysis of all samples indicated that 4-CQA was the predominant isomer in both sweet potato cultivars. Significant variations of p-coumaroyl quinic acids, feruloyl quinic acids, dicaffeoyl quinic acids, and tricaffeoyl quinic acid were identified, whereas the mono-caffeoyl quinic acids did not vary when the two sweet potato varieties were compared. Compared to conventional sweet potatoes, vegetable sweet potatoes exhibit a high negative correlation between 4-CQA and 5-pCoQA, while showing a high positive correlation between 3,5-CQA and 3-pCoQA. A series of principal component analyses (PCA) using CGA isomers enables a clear differentiation between vine tips derived from vegetable and conventional sweet potatoes. The model of linear discriminant analysis, based on the characteristic CGA, achieved a 100% accuracy rate in distinguishing between vegetable and conventional sweet potatoes. The high purity of sweet potato CGA (SCGA) exhibited potent anti-breast cancer activity. The results demonstrated that SCGA significantly suppressed the clonogenicity of MB231 and MCF7 cells, and impeded the migratory, invasive, and lung metastatic potential of MB231 cells.

## 1. Introduction

Chlorogenic acids (CGAs) are hydroxycinnamyl esters of quinic acid, which belongs to the class of plant secondary metabolites that are widely distributed throughout the plant kingdom. CGAs represent one of the most abundant phenolic compounds in our daily diet [1,2].

A sweet potato (*Ipomoea batatas* (L.) Lam) vine tip is a 15–20 cm segment of the growing point of the sweet potato vine, which comprises the stem, petiole, and leaves. The above components of sweet potato contain considerable phenolic compounds, and more than 70% of those phenolic compounds are composed of CGAs and their derivative compounds [1]. A vegetable sweet potato is a kind of sweet potato that is specially cultivated by collecting young shoot tips. Compared to conventional sweet potatoes, the aboveground portion of this variety exhibits vigorous growth, yet its tuber yield is notably low. The most desirable characteristics of vegetable sweet potatoes include its tender leaf tips with no or extremely little pubescence, and excellent eating quality relative to the regular variety [3]. However, the lack of current information on the CGAs of sweet potatoes seriously impedes their popularization and application. A comparison of CGA isomers in vegetable sweet potatoes and conventional sweet potatoes has not been previously documented.

Isomerism dominates the structural chemistry of CGA. The four non-equivalent hydroxyl substituents in quinic acid can generate four regioisomeric CGAs, and, for derivatives with multiple substitutions, additional regioisomeric structures may arise [4]. Coffee, as the primary dietary source of chlorogenic acid, yields 45 to 50 distinct CGA derivatives in Arabica coffee and up to 80 to 90 CGA derivatives in its close relative Robusta coffee [5]. The diversity and abundance of CGA isoforms are influenced by both intrinsic factors, such as genotype and species, and extrinsic factors, including source, harvest season, pretreatment method, and subsequent processing steps [6,7,8,9]. To explore these disparities, various techniques have been employed for the quantitative and qualitative analysis of natural plant CGA isomers, including near infrared spectroscopy [10], nuclear magnetic resonance spectroscopy [11], high-resolution mass spectrometry, and most notably high-performance liquid chromatography-mass spectrometry (HPLC-MS) [12]. In addition, multivariate statistical techniques such as CGA isomer ratio analysis, Pearson correlation coefficient analysis, and most importantly, principal component analysis (PCA) are employed to handle the vast amount of stoichiometric data obtained from these experimental methods [13].

In this study, we conducted statistical analysis on vine tip CGAs of different varieties of vegetable sweet potato (Shulv No. 1, Fu No. 18, Tainnong No. 71) and conventional sweet potato (Xu Zishu No. 8, Xu Shu No. 32, Xu Shu No. 29), selected based on their geographic diversity, yield potential, and availability for sourcing purposes. Highly purified CGA extracts were obtained through optimized extraction and purification methods. We conducted a comparison of the CGA profiles and CGA isoform ratios among the selected sweet potatoes, followed by PCA to visually represent the differences between various types of sweet potato. The scoring plots revealed specific markers that were responsible for significant variations observed in the sweet potato samples. Furthermore, we augmented the statistical analysis by incorporating the dominant isomers as focal points for examination of *p*-values and Pearson correlation coefficient.

## 2. Materials and Methods

### 2.1. Sample Preparation

The vine tips from six sweet potato cultivars, originating from the Institute of Sweet Potato of the Chinese Academy of Agricultural Sciences (117.305231 E, 34.283887 N, Xuzhou, China), were divided into vegetable sweet potato (Shulv No. 1, Fu No. 18, Tainnong No. 71) and conventional sweet potato (Xu Zishu No. 8, Xu Shu No. 32, Xu Shu No. 29). All cultivars were planted at the experimental farm at the Institute of Sweet Potato at the Chinese Academy of Agricultural Sciences in mid-March 2020, according to standard production procedures. The mean monthly temperature of 2020 was as follows: March 6–16 °C, April 9–21 °C, May 17–28 °C, June 21–30 °C, July 22–31 °C, and August 25–32 °C. The 15–25 cm long parts of the sweet potato vine tips were harvested at the end of August. The fresh sweet potato vine tips were freeze-dried, and then ground to a fine powder. The samples were sifted through 80 mesh sieves and stored at −20 °C for further use.

### 2.2. Optimization of Process Parameters for CGA Extraction from Sweet Potato Vine Tips

CGA refers to an organic-acid vinegar that comprises a carboxyl group and catechol hydroxyl group. It is soluble in alcohol and acetone, and has great solubility in hot water [12]. Combined with the consequent separation process, ethanol aqueous solution was selected as the extraction agent [1,4]. According to the results of laboratory optimization of extraction conditions (unpublished data), the above samples were mixed with a 50% ethanol/water extract at pH 3 in a ratio of 1:12 and extracted for 120 min at 68 °C using a constant temperature shaker. After filtration, crude CGA extracts were obtained.

### 2.3. Flocculation Precipitation and Decolorization

Before use, 1% acetic acid solution was prepared into 1% (*w*/*w*) chitosan colloid as a clarifying agent. The prepared chitosan clarifiers of 0.04% (chitosan weight, *w*/*w*) were introduced to the sample extract, and the samples were stirred at 60 °C for 30 min. The extraction solution with chitosan clarifier was kept at an ambient temperature for 1 h [14]. The centrifuged and filtered samples were subsequently stored at 4 °C.

Decolorization with activated carbon: The method was optimized based on a previous report [15,16,17]. 1% (*w*/*v*) activated carbon was added to the preheated extract at 60 °C, and decolorization was carried out with constant temperature stirring for 60 min. The resulting sample was filtered, concentrated by rotary evaporation to remove ethanol, and stored at 4 °C.

### 2.4. CGA Content Assay

Paper chromatography-UV spectrophotometry is one of the most rapid and accurate methods for the detection of active components of extracts in routine laboratories. [18]. Briefly, precision weigh 50 mg chlorogenic acid standard, then use ethanol to dissolve the volume to 10 mL. The resulting concentration is 5 mg/mL. Draw 5, 10, 15, 25, 30, 35, 40, 45, and 50 μL of the sample into a vial with a capacity of up to 10 mL and measure the absorbance at a wavelength of 330 nm. The standard curve regression equation was established with concentration as the ordinate and absorbance as the abscissa:Y = 0.468 × X + 0.004, R^2^ = 0.9895(1)

X for the measured absorbance,

Y for the chlorogenic acid content (mg).

The CGA-extracted solution was dribbled onto the chromatographic filter paper using a microinjector and dried using a hot air blower. The filter paper was expanded by 10 cm in a chromatography cylinder using distilled water. After air drying, the filter paper was positioned under a 365 nm UV lamp, and the chromatographic paper at the corresponding spots was cut out and extracted with 10 mL 95% ethanol by ultrasonic vibration for 15 min. The absorbance of the supernatant was detected at a wavelength of 328 nm, and the chlorogenic acid content was calculated according to the standard curve.

### 2.5. Purification of CGA Using NKA-9 Macroporous Adsorption Resin

A total of 400 mL of the crude CGA solution (2.0 mg/mL, pH 3.0) flows through the NKA-9 macroporous adsorption resin (M0032, Solarbio, Beijing, China) column at a flow rate of 1.0 BV/h (BV = 100 mL). When the adsorption reached equilibrium, the column was washed with 300 mL distilled water, and then eluted with ethanol solution (50%, 1.0 VB/h). The 5 BV samples can make the NKA-9 resin column reach adsorption equilibrium, and 3.5 BV volume eluate can completely desorb CGA from the saturated resin column. The desorption solution was concentrated via rotary evaporation, followed by freeze-drying, and storage at −80 °C.

### 2.6. Animals and Cell Experimental Protocol

#### 2.6.1. Migration and Invasion Experiments

Michigan Cancer Foundation-7 (MCF7) cells and Metastatic Breast-231 (MB231) cells were purchased from the Cell Bank of the Chinese Academy of Sciences Type Culture Collection. The cells were cultured in DMEM (BC-M-005, Bio-channel, Nanjing, China) containing 10% FBS (FSP500, ExCell Bio, Guangzhou, China) and incubated at 37 °C in a 5% CO_2_.

In the invasion experiment, the Transwell chamber was coated with Matrigel (AC-M082706, ACROBiosystems, Beijing, China), which was placed in a 24-well plate and incubated for 4 h to solidify. However, no coating was required for the migration experiment. The digested cells were resuspended in serum-free medium, counted; then, 40,000 cells were added to each chamber for the migration experiment, and 80,000 cells were added to each chamber for the invasion experiment. A total of 600 μL of medium containing 20% serum was added to the 24-well plate, followed by adding 200 μL of cell suspension to each cell chamber. After a migration period of 12 h or an invasion period of 24 h, the cells were transferred to a new well plate containing paraformaldehyde fixing solution (800 μL) and fixed at 4 °C for 30 min. The fixing solution was washed off with PBS before staining with crystal violet dyeing solution (800 μL) at room temperature for another 30 min. The Transwell chambers were washed with PBS and any remaining membrane cells on top of the upper compartment were wiped off using a cotton swab. The Transwell was air-dried, meticulously examined, and documented under a microscope, followed by a quantification of the obtained results.

#### 2.6.2. Mouse Model for Human Breast Cancer Lung Metastasis

Six-week-old female Balb/c-Nude (SCXK (Jing) 2021-0006) mice, obtained from Beijing Vital River Laboratory Animal Technology Co., Ltd. (Beijing, China), were acclimatized for one week before being randomly assigned to four groups (*n* = 10): Control and SCGA. Subsequently, each nude mouse received a tail vein injection of 2 × 10^5^ MB231 cells, while the SCGA group was orally administered 50 mg/kg/d CGA; the control group received an equivalent volume of conventional saline. The animals were housed at a temperature of 24 °C with 50% humidity and subjected to alternating cycles of light and dark for 12 h daily (lighting provided between 6:00 and 18:00). All animal experiments adhered to the protocols approved by the Institutional Animal Care and Use Committee of Xuzhou Medical University (IACUC Issue No. 202009W021 and Appl. Date: 15 September 2020).

### 2.7. High Resolution LC-MS^n^

Separation was attained using a 2.1 × 100 mm i.d. (Waters, ACQUITY UPLC HSS T3). The solvent A was 0.4% formic acid aqueous solution, and the solvent B was methanol. The total flow rate was set to 500 μL/min. The mobile phase gradient is as follows: linear increase from 10% B to 70% B within 45 min, isodegree for the following 10 min, return to 10% B for rebalancing at 75 min, and then 10 min isocratic for re-equilibration. The LC equipment (Ulti Mate 3000, THERMO, Waltham, MA, USA) included a binary pump, an auto-sampler, and a DAD detector (recording at 320 and 254 nm and scanning from 200 to 600 nm). This was interfaced with Triple TOF 5600 mass spectrometers (AB SCIEX, Singapore) operating in full scan. ESI ion source parameters were set according to the literature [19]. The contents of 12 CGAs were expressed as microgram CGA per gram sample (mg/g sample) on dry weight basis. The regressive equations and correlation coefficients for CGAs standards are provided in Appendix A.

### 2.8. Statistical Evaluation

The mean, minimum, and maximum values of all the vegetable and conventional sweet potato CGA were computed as shown in Appendix A. These values were further used in calculating the skewness degree, to determine the symmetry level of the data distribution, and kurtosis, to measure the data tailing degree relative to conventional distribution towards determining the outliers’ level in the data set we evaluated. In addition, the *p*-values were computed, and principal component analysis (PCA) was carried out using the Orange 3.0 data mining software tool as previously reported for the data set with diverse variables degree [13]. Linear discriminant analysis was performed using SPSS 24.0 software.

## 3. Results and Discussion

### 3.1. CGA Extraction and Purification of Sweet Potato Vine Tips

The outline features of the vine tips of vegetable sweet potatoes (Shulv No. 1, Fu No. 18, Tainn No. 71) and conventional sweet potatoes (Xu Zishu No. 8, Xu Shu No. 32, Xu Shu No. 29) are depicted in Appendix A. Compared to conventional sweet potatoes, the vines of vegetable sweet potatoes are noticeably softer and devoid of fluff. Furthermore, the vine growth rate of vegetable sweet potatoes is significantly higher (Figure 1 and Appendix A).

The crude extract of sweet potato vine tips contains a diverse range of impurities, pigments, tannins, proteins, colloids, and sugars. Direct utilization of macroporous adsorption resin for the separation and purification of the crude extract results in impairment of the resin’s microporous structure due to these impurities, leading to a significant reduction in the adsorption efficiency of macroporous adsorption resin towards CGA. Therefore, it is imperative to conduct an initial process aimed at removing impurities from the CGA crude extract.

Chitosan disperses in an acidic aqueous solution to form a polymer solution and is a commonly used flocculant [20]. The chitosan clarifier exhibits distinctive characteristics: Firstly, the presence of amino and carboxyl groups within the chelation and precipitation of heavy metal ions, thereby facilitating the removal of residual heavy metals from solutions. Secondly, its elongated chain structure exerts a robust “bridging” effect that allows for the adsorption of multiple particles onto a single molecule, leading to aggregation and subsequent precipitation of suspended particles exhibiting Stoke’s settling in the solution. Thirdly, the positively charged amino group in chitosan facilitates binding with negatively charged proteins, amino acids, and other compound molecules. Through the above three actions, chitosan solution can produce an effective clarifying effect on a specific dispersion system. CGA molecules do not contain heavy metal ions but do contain a multitude of hydroxyl groups with positive charges [21]. From the function of chitosan clarifier, it is clear that it has no specific adsorption capacity for the polar molecule CGA.

The pigment will also contaminate the resin and reduce its adsorption capacity for CGA. The microporous structure of activated carbon exhibits exceptional adsorption capabilities towards pigment molecular pollutants, rendering it a highly efficient decolorization method [15,16,17]. Activated carbon exhibits a limited capacity for CGA molecule adsorption, with relatively weak binding affinity. By optimizing the dosage, duration, and temperature of the decolorizing agent, it is possible to significantly enhance the decolorization efficiency of the extract while minimizing the loss rate of CGA.

The crude extract was dynamically adsorbed and desorbed using a resin column. We concentrated the eluate at 45 °C by rotary evaporator, and then freeze dried it. The extraction of over 5 g of purified product was achieved from all 100 g dry-powder samples of sweet potato vine tips. The purity of the purified samples was determined using Paper chromatography-UV spectrophotometry, revealing that all samples exhibited a purity exceeding 85% (Table 1).

### 3.2. Qualitative and Quantitative Analysis of CGA in Sweet Potato Vine Tips by LC-MS

Studies have shown that the proportion of CGA isomers in different types of coffee beans is significantly different, and some of the above CGA isomer compositions can serve as a sign to distinguish different sources of coffee [22]. The percentage composition of different CGAs in the plant is affected by the species, maturity, origin, cultivation, and climatic conditions, and the method of handling (drying or lyophilization) [23]. Accordingly, the LC-MS method was used to perform qualitative and quantitative analysis of CGA and its isomers in sweet potato vine tips to determine the proportion of CGA isomers of three conventional and three vegetable sweet potato varieties. The results showed that the same CGA isomer species were detected in the vine tips of vegetable and conventional sweet potatoes, but the content and ratio were significantly different. All data for chlorogenic acids presented in this manuscript use the recommended IUPAC numbering system [24]. Chemical structures of key CGAs are depicted in Appendix A, accompanied by their respective details presented in Table 2. The chromatogram provided in Figure 1 highlights the identified CGAs in this study. Subsequently, it is imperative to conduct further analysis of the isomeric composition of CGA and compare rattan tip SCGA derived from vegetable sweet potatoes with that obtained from conventional sweet potatoes.

#### 3.2.1. Mono-Caffeoylquinic Acids (CQA) and di-Caffeoylquinic Acids (diCQA)

The absolute values of the mono-caffeoyl quinic acids (CQAs) were obtained and represented in the box plot (Figure 2A and Appendix A) with a typical chromatogram (Figure 1). In this study, three isomers of mono caffeoylquinic acid including 3-CQA, 4-CQA and 5-CQA were detected. The analytical results showed that 4-CQA was the dominant isomer in all samples when compared with 3-CQA and 5-CQA. The findings of previous studies differ from our results, as they indicated that 5-CQA was the primary constituent of CGA in sweet potatoes [25]. As depicted in Figure 2A, a significant disparity was observed in the concentration of 4-CQA (vegetable: 24.57 ± 6.74 mg/g; conventional: 28.05 ± 4.41 mg/g) between the vegetable and conventional sweet potato vine tips, whereas the levels of 3-CQA (vegetable: 2.34 ± 0.33 mg/g; conventional: 2.21 ± 0.32 mg/g) and 5-CQA (vegetable: 2.95 ± 0.67 mg/g; conventional: 2.52 ± 0.90 mg/g) were found to be insignificantly different. The conventional sweet potato vine tips had higher 4-CQA content than the vegetable sweet potato vine tips. Existing research has suggested that the main CGA components of coffee beans are primarily 5-CQA [26]. The major CGA component of honeysuckle, dandelion, and *eucommia* leaves is 3-CQA, which is different from that of green coffee beans [27].

All three dicaffeoyl quinic acids were detected in a typical chromatogram. (Figure 1) and four derivatives were quantified with the resulting box plot in Figure 2B. Previous studies have demonstrated that sweet potato leaves contain substantial amounts of all three isomers of DiCQA. However, the DiCQA of different sweet potato species was not compared [27,28]. In this study, a significant disparity was observed in the total content of dicoyl-quinic acid (DiCQA) between vegetable and conventional sweet potato vine tips within CGA. Furthermore, the levels of 3,4-DiCQA and 3,5-DiCQA were significantly higher in vegetable sweet potato vine tips compared to those found in conventional sweet potato vine tips. Notably, only the concentration of 4,5-DiCQA did not exhibit any significant difference when compared to that of 3,4-DiCQA and 3,5-DiCQA. From the plot it can be seen that the vegetable sweet potato vine tips that were DiCQA quantified contained 4.58 ± 1.87 mg/g of 3,4-DiCQA, 11.81 ± 3.83 mg/g of 3,5-DiCQA, and 1.98 ± 0.80 mg/g of 4,5-DiCQA; while conventional sweet potato vine tips contained 2.64 ± 0.94 mg/g of 3,4-DiCQA, 2.69 ± 1.15 mg/g of 3,5-DiCQA, and 0.87 ± 0.38 mg/g of 4,5-DiCQA.

#### 3.2.2. Feruloyl Quinic Acids (FQA), –p-Coumaroyl Quinic Acids (pCoQA)

Figure 2C displays the quantitative results of the FQAs. The plot illustrates that the vegetable sweet potato contained 0.53 ± 0.14 mg/g of 5-FQA, 2.39 ± 0.87 mg/g of 4-FQA, and 0.59 ± 0.15 mg/g of 3-FQA. Additionally, the conventional sweet potato contained 1.00 ± 0.21 mg/g of 5-FQA, while 4-FQA and 3-FQA were present at concentrations of 5.11 ± 1.52 mg/g and 0.92 ± 0.12 mg/g, respectively. The above findings suggest that the levels of 4-FQA were significantly elevated in both sweet potato species when compared to those of 3-FQA and 5-FQA. Furthermore, in the conventional sweet potato samples, FQA was more present than in the vegetable sweet potato samples. 5-FQA in the coffee extract was significantly higher than 3-FQA and 4-FQA [26]; not consistent with CGA extract from sweet potato vine tips.

The pCoQA acid components of the vegetable and regular sweet potato vine tip samples were quantified and are represented in Figure 2D. The analysis results showed that the total pCoQA content of conventional sweet potato samples was significantly higher than that of the vegetable sweet potato samples. Compared to 5-pCoQA, the predominant isomers in conventional sweet potato samples were 3-pCoQA and 4-pCoQA, which were significantly higher than those found in the vegetable sweet potato samples. Unlike sweet potato, 4-pCoQA is the most dominant regional isoform in green coffee extract [26] compared with 3-pCoQA and 5-pCoQA.

### 3.3. Inspection of Characteristic Isomer Ratio Parameters by Comparing Vegetable and Conventional Samples CGA Quantitative Data

Isomer proportion and composition can be useful as a means of distinguishing between species, and it is well documented that roseaceae such as apples, apricots, pears, plums, and peaches preferentially produce three substituted CGAs, whereas in most other plant families, the 5-isomer predominates [21,29]. Studies have shown that the ratio of different CGA content in coffee is influenced by a number of factors, including the variety of beans, maturity, country of origin, cultivation and climatic conditions, the presence of defective beans, and the method of processing wet or dry, washed or unwashed, beans after harvest [23,30]. Chlorogenic acid isomer ratios have been demonstrated to differentiate between various sources of coffee [22]. Therefore, we have decided to further investigate these parameters by calculating isomer ratios based on the available quantification results. The box plot in Figure 3 displays the ratios of CQA, DiCQA, FQA, and pCoQA constituents present in the sample.

The result reflects the conventional samples’ CQA isomers ratios; mean 4/5-CQA appears to be higher with 12.08 compared to vegetable samples with 8.69 (Figure 3A). The results of the DiCQA isomers showed that the average ratio value of 3,4/4,5 for the vegetable sweet potato was 6.45; significantly higher than that of the conventional sweet potato samples at 3.32 (Figure 3B). The proportions of FQA and CQA isomers were comparable (Figure 3C), with an average 4/5 ratio for the conventional sweet potato at 5.52, which was significantly higher than that of the vegetable sweet potato (4.0). Furthermore, the pCoQA ratios were calculated and plotted as indicated in Figure 3D. The ratios of 3/5 and 4/5 for the conventional sweet potato were found to be significantly higher than those for vegetable sweet potato, with values of 10.33 and 12.90, respectively, compared to only 1.77 and 0.71 for the latter.

### 3.4. Statistical Analysis Using p-Values and Pearson Correlation Coefficient

Close examination of the data distribution revealed that all sweet potato CGAs exhibited non-Gaussian characteristics. Similarly, an analysis of variance homogeneity clearly indicated heteroscedasticity in all analyte datasets studied (Appendix A). This is consistent with the distribution results of some CGAs in coffee [26]. This clearly exemplifies the pivotal role of CGAs in plant defense, as plants experiencing stress may exhibit an augmented demand for CGAs [26,31]. Therefore, it is imperative to conduct a comprehensive analysis of the disparities observed among the 12 detected CGAs.

Appendix A and Figure 4A,B presents the *p*-values (*p* < 0.05) and Pearson correlation coefficients in a statistical manner, utilizing paired two-sample t-tests to compare the significance between the 12 CGA isoforms. Since the 4-CQA and 3,5-CQA isomers were typically the most prevalent of the CGAs found in sweet potato vine tips, it is crucial to consider its correlation with other CGAs present in these tips. Based on the data presented in Appendix A and Figure 4A,B, the multiple sets of chlorogenic acid isomer *p* values for both vegetable sweet potato and conventional sweet potato exhibited varying levels of statistical significance. This suggests that a comparison between 4-CQA, 3,5-CQA and other CGA isoforms is statistically significant. Furthermore, we have observed that the combination of statistically significant CGA isomers in vegetable sweet potato purposes is similar to that found in the conventional sweet potato. This suggests that both types possess a comparable number of indicators utilized to differentiate between vegetable and conventional sweet potato vine tips.

The Pearson correlation coefficients of 4-CQA and 3,5-CQA compared with other isomers are shown in Figure 4A,B, which shows that there are some differences in the correlation. Most significantly, the 4-CQA vegetable sweet potato, when compared with 5-pCoQA vegetable sweet potato, shows high negative correlation with a coefficient of −0.8742 which is closer to −1 compared with the conventional sweet potato which has a coefficient of 0.2587. Similarly, the 3,5-CQA, when compared with 4-pCoQA, has 0.831 for the vegetable sweet potato compared with the conventional sweet potato which has −0.2681. It can be inferred that 5-pCoQA and 4-pCoQA can be used as biomarkers to accurately distinguish vegetable sweet potatoes from conventional sweet potatoes.

### 3.5. Principal Component Analysis and Linear Discriminant Analysis Model

Through the analysis of the chlorogenic acid isomer content and Pearson coefficients of the two sweet potato vine tips, significant differences in chlorogenic acid were observed. To further investigate this finding, we conducted a series of principal component analyses (PCAs) to compare the CGA content between the two varieties of sweet potatoes. Additionally, we built a model based on linear discriminant analysis (LDA) to allow clear differentiation between them.

With the CGA content data sets recorded, PCA was chosen as an unsupervised method for statistical sample comparison. PCA identifies the principal components of a dataset by determining linear combinations of its original variables that capture the maximum amount of variance. Visualizing the dataset in terms of principal components results in a score plot, which facilitates sample grouping based on variance, and indicates the degree of variance based on the distance between two points. In addition, a loading plot can be used to visually represent the contribution of each compound to the principal components [32]. The greater the absolute value of a specific compound on one of the axes defined by the principal components, the stronger its impact on variance (and thus separation) along the corresponding principal component in the score plot [32]. We performed PCA analysis on all samples, and the resulting score plot and loading plot are shown in Figure 5A,B. A clear segregation between vegetable and conventional sweet potato is evident in the score plot, with an explained variance of 75% in PC1 and PC2. The compounds that contribute the most to the loading plot are annotated in their corresponding positions. Combined with the loading plot, it is evident that the vegetable sweet potato exhibits a strong correlation with 3,4-CQA, 3,5-CQA, and 4,5-CQA; whereas, the conventional sweet potato displays a high association with 4-CQA, 3-FQA, 4-FQA, 5-FQA, and 3-pCoQA.

Badmos et al. identified 15 chlorogenic acids as the primary distinguishing characteristics and developed an LDA model capable of accurately classifying both Robusta and Arabica coffee varieties with a 100% success rate [26]. Based on the Fisher linear discriminant model of CGAs, 12 isomers (3-CQA,4-CQA, 5-CQA 3,4-DiCQA, 3,5-DiCQA, 4,5-DiCQA, 3-FQA, 4-FQA, 5-FQA, 3-pCoQA, 4-pCoQA, and 5-pCoQA) that made significant contributions to principal component analysis were selected as indicators for the binary Fisher linear discriminant analysis. The coefficients obtained for discriminant function and classification function are shown in Table 3. The model achieved 100% accuracy in distinguishing between vegetable sweet potatoes and conventional sweet potatoes. To validate the model, we tested six additional samples (three vegetable and three conventional, the acquisition conditions of test samples were consistent with those of modeling samples), all of which were correctly classified with 100% accuracy.

### 3.6. Purified Sweet Potato CGA (SCGA) Bioactivity Assay

Several studies have shown that chlorogenic acid has anti-tumor properties. The compound CGA exerts its inhibitory effect on DNA methylation through a non-competitive mechanism, thereby significantly suppressing the proliferation of MCF7 breast cancer cells and inducing cell cycle arrest in the G0/G1 phase [33]. Jiang et al. discovered that, under alkaline conditions, chlorogenic acid exhibits pro-oxidizing properties, thereby inducing the generation of large DNA fragments in tumor cells and subsequently leading to nucleoagglutination; ultimately inhibiting tumor cell proliferation [34]. The growth of human leukemia K562/ADM cells can be inhibited by CGA, which also effectively reverses their multi-drug resistance (MDR). This is primarily achieved through the inhibition of the PI3K/Akt signaling pathway, leading to the down-regulation of the expression of multi-drug-resistance associated albumin (MRP1) and P-glycoprotein (P-gp) genes [35]. The expression of HIF-1 and VEGF proteins in lung cancer cells is effectively reduced by CGA, thereby exerting a significant inhibitory effect on tumor angiogenesis [36].

To investigate the impact of SCGA on breast cancer cell proliferation, MB231 and MCF7 cells were treated with SCGA, and clonogenicity was assessed using a clonogenic assay. The results demonstrated a significant inhibition of breast cancer cell clonogenesis upon treatment with SCGA (Figure 6A). Furthermore, the effect of SCGA treatment on the migration and invasion abilities of MB231 cells is depicted in Figure 6B, revealing a substantial suppression by SCGA.

The lung metastasis model of breast cancer was established by injecting MB-231 cells into the tail vein of nude mice. Concurrently, intragastric administration of SCGA (50 mg/Kg/d) was performed in the nude mice. After two months, the lungs were dissected and examined, revealing a significant reduction in lung metastases in the SCGA group compared to the control group (Figure 6C). Statistical analysis demonstrated a significant difference in the number of lung metastases following SCGA treatment (Figure 6D). Furthermore, survival time analysis indicated an extended survival period in the SCGA treatment group (Figure 6E). In the aforementioned experiments, SCGA, a naturally derived compound extracted and purified from the stem tip of sweet potatoes, exhibited potent anti-breast tumor activity.

## 4. Conclusions

In this study, CGA was extracted from sweet potato vine tips using the alcohol extraction method. Subsequently, purification steps including flocculation precipitation, activated carbon decolorization, and NKA-9 macroporous adsorption resin were employed. The resulting purified product yielded over 5 g of CGA from every 100 g dry-powder sample of sweet potato vine tips, with a purity exceeding 85%. LC-MS analysis revealed the presence of various CGAs in the purified extract obtained from sweet potato vine tips, including CQA, DiCQA, FQA, and pCoQA. Among these isomers, 4-CQA and 3,5-CQA were found to be predominant in all samples compared to other CGAs. A series of principal component analyses (PCA) using 12 CGAs isomers (3-CQA,4-CQA, 5-CQA 3,4-DiCQA, 3,5-DiCQA, 4,5-DiCQA, 3-FQA, 4-FQA, 5-FQA, 3-pCoQA, 4-pCoQA, and 5-pCoQA) enables a clear differentiation between vine tips derived from vegetable and conventional sweet potatoes. The model of LDA, based on the characteristic 12 CGAs isomers, achieved a 100% accuracy rate in distinguishing between vegetable and conventional sweet potatoes. SCGA exhibited significant anti-breast tumor activity as evidenced by clonal formation experiments, migration invasion experiments, and lung metastasis experiments, thus highlighting its potential as a natural active product for further research and development.

## Figures and Tables

**Figure 1 foods-12-03910-f001:**
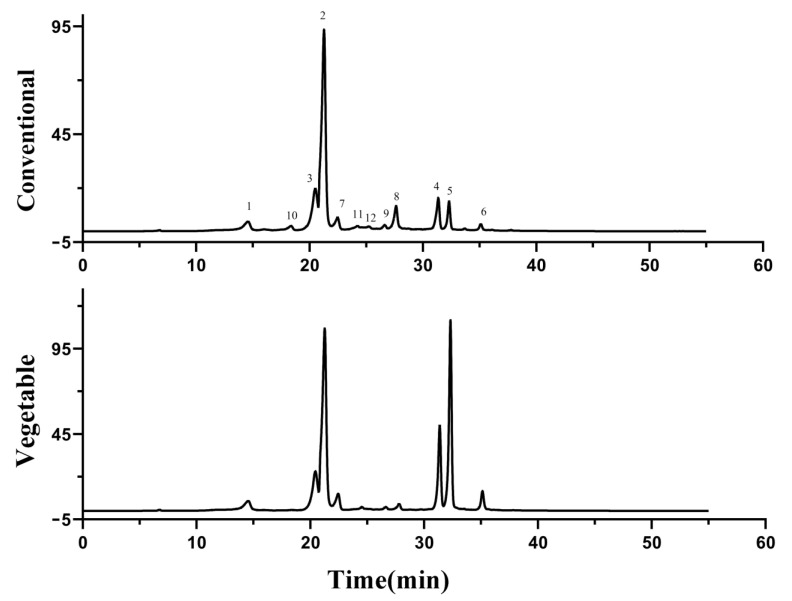
The chromatograms representing vegetable and conventional sweet potato vine tips CGA. 1: 3-CQA, 2: 5-CQA, 3: 4-CQA, 4: 3,4-diCQA, 5: 3,5-diCQA, 6: 4,6-diCQA, 7: 3-FQA, 8: 4-FQA, 9: 5-FQA, 10: 3-pCoQA, 11: 4-pCoQA, 12: 5-pCoQA.

**Figure 2 foods-12-03910-f002:**
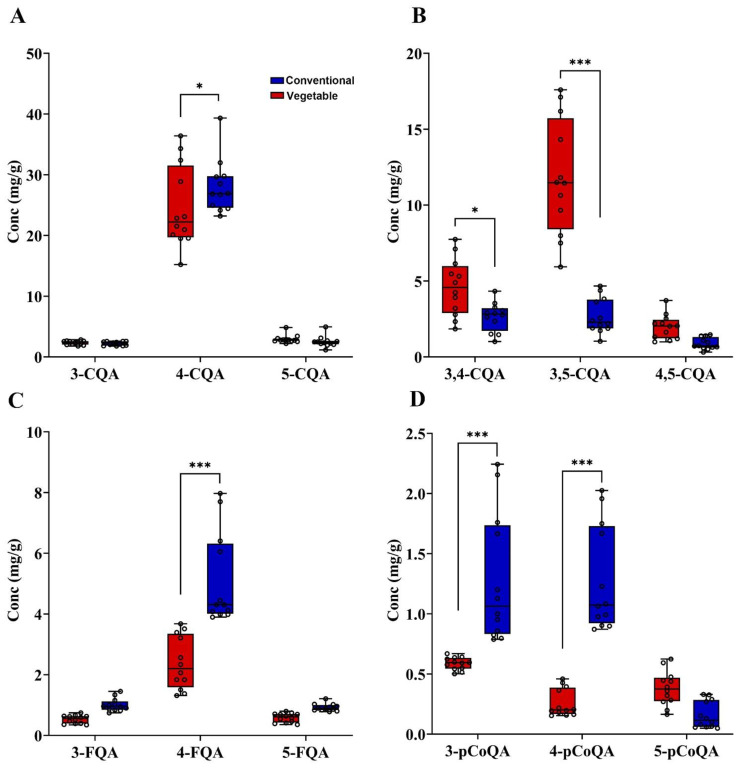
Content of CGA from sweet potato vine tips. (**A**) Comparison of mono-caffeoylquinic acids content (mg/g). (**B**) Comparison of di-caffeoylquinic acids content (mg/g). (**C**) Comparison of mono-feruloylquinic acids content (mg/g). (**D**) Comparison of para-Coumaroylquinic acid relative content. * *p* < 0.05, *** *p* < 0.001.

**Figure 3 foods-12-03910-f003:**
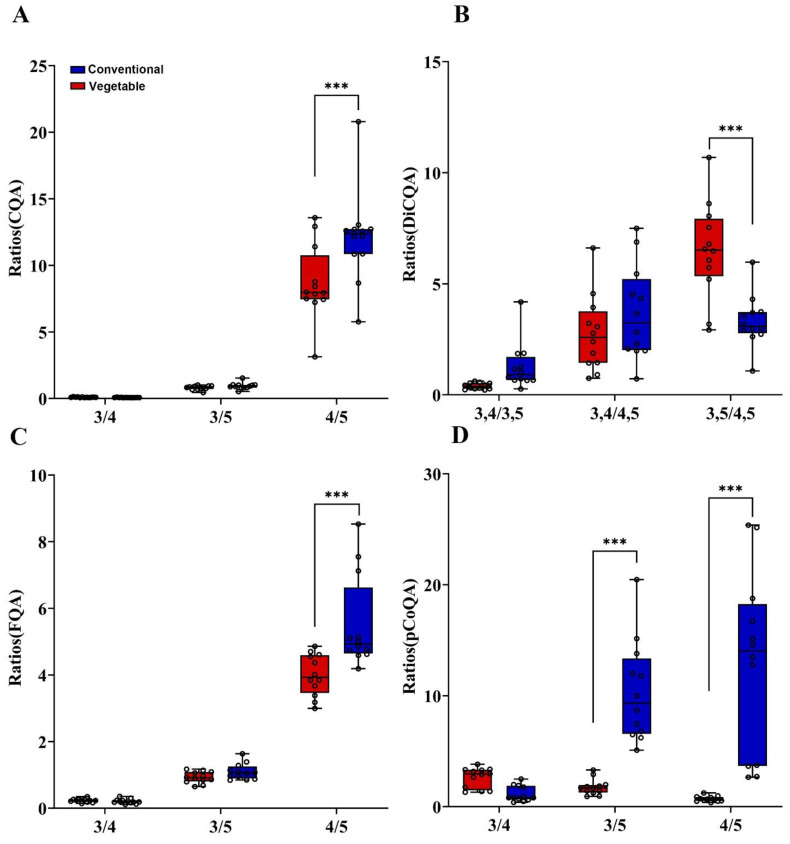
Box-plot of regio-isomer ratios of the CQA, DiCQA, FQA and pCoQA constituents of conventional (blue) and vegetable (red) sweet potato vine tips. Mean (*n* = 12) and expressed in mg/g. (**A**) The ratios of 3-CQA to 4-CQA, 3-CQA to 5-CQA, 4-CQA to 5-CQA; (**B**) The ratios of 3,4-DiCQA to 3,5-DiCQA, 3,4-DiCQA to 4,5-DiCQA and 3,5-DiCQA to 4,5-DiCQA; (**C**) The ratios for the FQA; (**D**) The ratios for the pCoQA. *** *p* < 0.001.

**Figure 4 foods-12-03910-f004:**
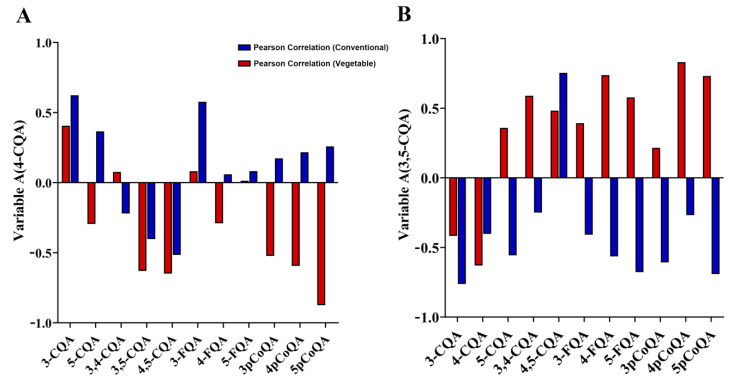
Pearson correlation coefficient with a fixed substituent showing statistical significance of the comparison between individual 4-CQA (**A**), 3,5-CQA (**B**) and other isomers in vegetable and conventional sweet potato samples.

**Figure 5 foods-12-03910-f005:**
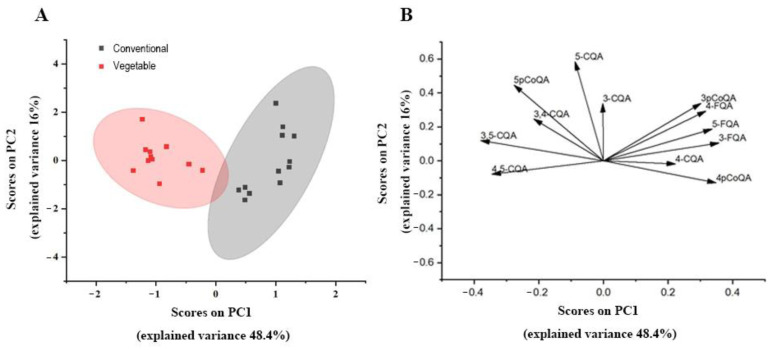
Principal component analysis (PCA) of CGA isomers from vegetable and conventional sweet potato samples. (**A**) Score plot of the PCA showing a clear distinction between vegetable and conventional samples. (**B**)Annotated loading plot of the PCA.

**Figure 6 foods-12-03910-f006:**
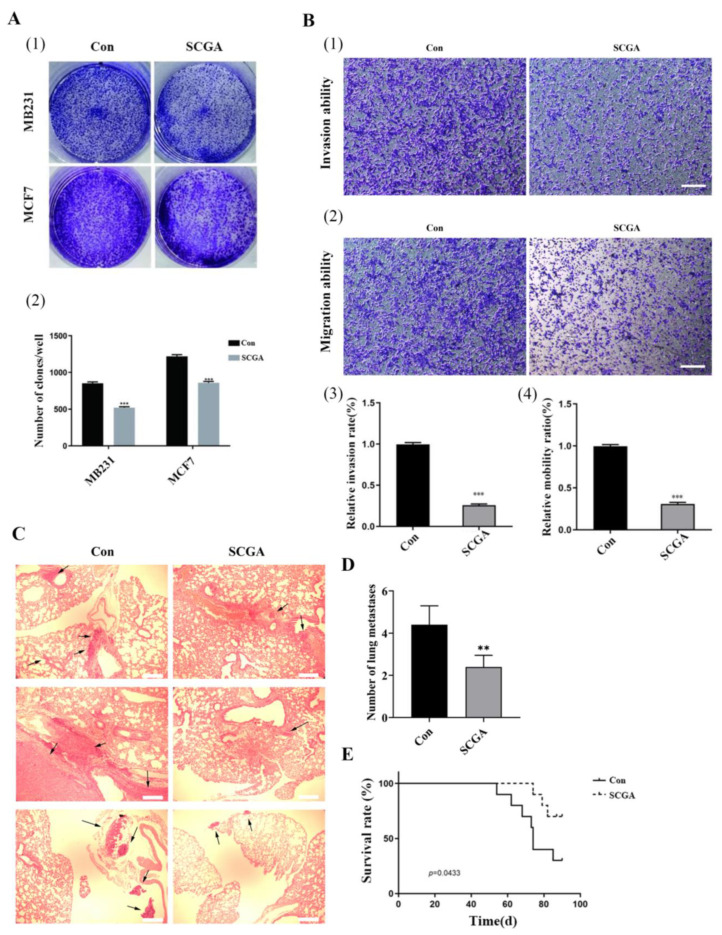
SCGA inhibits the activity of breast tumor cells. (**A**) Effect of SCGA on clonal formation ability of breast tumor cells. (**B**) Effect of SCGA on migration and invasion ability of breast tumor cells. (**C**) H&E staining results of lung tissue sections in nude mouse lung metastasis model. scale bar = 50 μm. The arrow points to lung metastatic breast tumor cells. (**D**) SCGA reduces the number of lung metastases from breast cancer in nude mice. The results are presented as the means ± SDs of three independent experiments. (**E**) Survival curve of nude mice with lung metastasis from breast cancer improved by SCGA. ** *p* < 0.01, *** *p* < 0.001.

**Table 1 foods-12-03910-t001:** Extraction quantity and purity of chlorogenic acid from sweet potato leaves of different varieties.

	Vegetable Sweet Potato			Conventional Sweet Potato		
	Shulv No. 1	Fu No. 18	Tainong No. 71	Xushu No. 32	Zishu No. 8	Xushu No. 29
Extraction yield (g/100 g)	4.90 ± 0.499	6.35 ± 0.45	6.07 ± 0.51	5.42 ± 0.49	5.04 ± 0.75	5.89 ± 0.55
Purity of CGA (%)	95.46 ± 2.37	85.79 ± 3.15	87.5 ± 4.14	85.21 ± 3.85	88.78 ± 4.57	91.87 ± 5.77

**Table 2 foods-12-03910-t002:** List of identified CGAs with their respective names and molecular formula.

No.	CGA Name	Abbreviation	Molecular Formula	Theor. *m*/*z* (M-H)
1	3-O-caffeoylquinic acid	3-CQA	C_16_H_18_O_9_	353.0878
2	4-O-caffeoylquinic acid	4-CQA	C_16_H_18_O_9_	353.0878
3	5-O-caffeoylquinic acid	5-CQA	C_16_H_18_O_9_	353.0878
4	3,4-di-O-caffeoylquinic acid	3,4-diCQA	C_25_H_24_O_12_	515.1195
5	3,5-di-O-caffeoylquinic acid	3,5-diCQA	C_25_H_24_O_12_	515.1195
6	4,5-di-O-caffeoylquinic acid	4,5-diCQA	C_25_H_24_O_12_	515.1195
7	3-O-feruloylquinic acid	3-FQA	C_17_H_20_O_9_	367.0929
8	4-O-feruloylquinic acid	4-FQA	C_17_H_20_O_9_	367.0929
9	5-O-feruloylquinic acid	5-FQA	C_17_H_20_O_9_	367.0929
10	3-O-p-coumaroylquinic acid	3-pCoQA	C_16_H_18_O_8_	337.0929
11	4-O-p-coumaroylquinic acid	4-pCoQA	C_16_H_18_O_8_	337.0929
12	5-O-p-coumaroylquinic acid	5-pCoQA	C_16_H_18_O_8_	337.0929

**Table 3 foods-12-03910-t003:** Discriminant and classification coefficients based on linear discriminant analysis of the 12 most predominant chlorogenic acids.

	Discriminant Function	Classification Function	
	1	Vegetable	Conventional
3-CQA	3.192	87.860	49.927
4-CQA	0.158	2.860	0.986
5-CQA	−0.351	−4.717	−0.546
3,4-CQA	0.503	6.623	0.652
3,5-CQA	−0.052	−1.672	−1.050
4,5-CQA	0.236	24.098	21.296
3-FQA	−5.725	−80.520	−12.487
4-FQA	0.365	15.690	11.347
5-FQA	−3.382	7.099	47.283
3-pCoQA	−5.746	−102.805	−34.524
4-pCoQA	−1.203	1.766	16.065
5-pCoQA	15.227	172.612	−8.325
Constant	−4.460	−162.887	−109.889

## Data Availability

The datasets of the current study are available from the corresponding author on reasonable request.

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
