# Peer review of "Composition and Bioactivity of Chlorogenic Acids in Vegetable and Conventional Sweet Potato Vine Tips"

_foods, 2023, doi:10.3390/foods12213910_

Round 1

Reviewer 1 Report

1. "Before use, 1% acetic acid solution was prepared into 1%(W/W) chitosan colloid as a clarifying agent." 

The clarifying agent could attracts active compounds from the extracts that you  have been prepared?

2. Row 126 "The extracted solution.." this solution is your purified extract?

3. Why you red absorbance of the solution obtained after spot was dissolved in water and you don't read the absorbance of the spot on the paper in the same way like calibration curve?

4. How you choose to use this SCGA dose in your animal experiment?

Author Response

Thank you for your letter and the reviewers’ comments concerning our manuscript entitled "Composition and bioactivity of chlorogenic acids in vegetable and conventional sweet potato vine tips" (ID: foods-2640769). 

We studied the reviewers’ comments carefully and tried to revise the manuscript according to the comments. The following are the responses and revisions we made in response to the reviewers’ questions and suggestions on an item-by-item basis. All the changes are marked in red. Point by point responses to the reviewers’ comments are listed below this letter.

Response to Reviewer 1:

Comments 1: "Before use, 1% acetic acid solution was prepared into 1%(W/W) chitosan colloid as a clarifying agent."

The clarifying agent could attracts active compounds from the extracts that you have been prepared?

Response:

Thank you for your comments. Flocculation precipitation separation technology refers to the intermolecular interaction with macromolecular substances such as protein pectin in the way of adsorption bridging and electric neutralization, so as to settle, remove the coarse particles in the solution, and achieve refining and impurity removal.

Chitosan disperses in acidic aqueous solution to form polymer solution and is a commonly used flocculant. Chitosan clarifier has unique characteristics: First, amino and carboxyl groups in the molecule can chelate and precipitate heavy metal ions, which can remove some heavy metal residues in the solution. Second, long chain molecules have a strong "bridging" effect, which can adsorb multiple particles on the same molecule, so that suspended particles with Stoker precipitation trend in the solution can aggregate and precipitate. Third, the amino group in chitosan has a positive charge and can be combined with negatively charged proteins and amino acids and other compound molecules. Through the above three actions, chitosan solution can produce an effective clarifying effect on a specific dispersion system.

Chlorogenic acid molecules do not contain heavy metal ions, and contain a multitude of hydroxyl groups with positive charges. From the function of chitosan clarifier, it has no specific adsorption capacity for polar molecule chlorogenic acid, and chlorogenic acid may be precipitated by binding with other macromolecules. Therefore, it is necessary to adjust the dosage and adsorption time to grasp the balance point between the clarification degree and the CGA loss rate, and achieve the best clarification conditions

We have added a discussion of the flocculation and precipitation process to the results and Discussion 3.1 to make it easier for readers to understand the principle and necessity of flocculation and precipitation in CGA extraction and purification process.

Comments 2: Row 126 "The extracted solution." this solution is your purified extract?

Response:

Thank you for your comments. "The extracted solution." It's the extract that I extract CGA from the tip of the sweet potato vine.

Comments 3: Why you red absorbance of the solution obtained after spot was dissolved in water and you don't read the absorbance of the spot on the paper in the same way like calibration curve?

Response:

Thank you for your comments. Compared with CGA standard liquid phase, CGA extract of sweet potato contains impurities. If detected directly by ultraviolet spectrophotometer, impurities similar to CGA absorption spectrum will affect the detection accuracy. Therefore, it is necessary to use paper chromatography to separate the high-purity CGA, cut the high-purity CGA from the filter paper and then dissolve it for detection, which can significantly improve the detection accuracy.

Comments 4: How you choose to use this SCGA dose in your animal experiment?

Response:

Thank you for your comments. The amount of SCGA was determined by literature review. Studies have shown that people can consume 1000mgCGA by drinking 4-5 cups of standard 250mL coffee per day.

Reviewer 2 Report

This manuscript compared the composition of chlorogenic acids (CGA), prepared the purified CGA extract, and mesured anti-breast cancer activity.

However, statistical results better be rearranged. In table 3, mean ± DS will be enough. Min, Max, Skewness, and Kurtosis are not necessary because figure 2 provide enough information.

In addition, table 4, figure 4, figure 5, and table 5 are somewhat repetitive. Please make them simple.

Please provide the source of standards.

Author Response

Response to Reviewer

Thank you for your letter and the reviewers’ comments concerning our manuscript entitled "Composition and bioactivity of chlorogenic acids in vegetable and conventional sweet potato vine tips" (ID: foods-2640769). 

We studied the reviewers’ comments carefully and tried to revise the manuscript according to the comments. The following are the responses and revisions we made in response to the reviewers’ questions and suggestions on an item-by-item basis. All the changes are marked in red. Point by point responses to the reviewers’ comments are listed below this letter.

Response to Reviewer 2:

Comments: However, statistical results better be rearranged. In table 3, mean ± DS will be enough. Min, Max, Skewness, and Kurtosis are not necessary because figure 2 provide enough information.

In addition, table 4, figure 4, figure 5, and table 5 are somewhat repetitive. Please make them simple.

Please provide the source of standards.

Response:

Thank you for your comments. We have removed Tables 3 and 4 from the original text and presented them as additional material.  Figure 5 shows the difference degree of CGA isomer content between vegetable sweet potato and traditional sweet potato as shown by PCA statistical analysis results, while Table 5 shows the equation coefficient of LDA model of CGA isomer content used to accurately distinguish vegetable sweet potato and traditional sweet potato. In addition, we have added all standard source information to the supplement Table S2.

Reviewer 3 Report

Comment and suggestion to authors:

Manuscript ID: foods-2640769

Type: Article

Titled:  Composition and bioactivity of chlorogenic acids in vegetable and conventional sweet potato vine tips

1)      As this study aim to investigate the composition and bioactivity of chlorogenic acids in vegetable sweet potato (Shulv No. 1, Fu No. 18, 80 Tainnong No. 71) and conventional sweet potato (Xu Zishu No.8, Xu Shu No.32, Xu Shu 81 No.29). So, the authors should add the photo of vegetable sweet potato compare with conventional sweet potato. In addition, their graphic diversity and habitat should be provided in the manuscript as well.

2)      Line 65-68: As the authors wrote in the manuscript that “In this study, we conducted statistical analysis on vine tip CGAs of different varieties of vegetable sweet potato (Shulv No. 1, Fu No. 18, Tainnong No. 71) and conventional sweet potato (Xu Zishu No.8, Xu Shu No.32, Xu Shu No.29), selected based on their geographic diversity, yield potential and availability for sourcing purposes.”. So, the additional table summarized the information about geographic diversity, yield potential and availability for sourcing purposes of vegetable vs. conventional sweet potato should be provided in the manuscript.

3)      In scientific name of this plant material should be provide in the introduction of the manuscript.

4)      In the figure 6 (B)Effect of SCGA on migration and invasion ability of breast tumor” should be revised. The higher resolution of figure and the clear bar scale should be added. The current bar scale of figure 6 (B) is very difficult to read.

5)      The greater number of another related published works should be added to discuss with the results from this current study.

6)      There are some spelling mistakes and grammatical error found in this manuscript.

Minor editing of English language required. There are some spelling mistakes and grammatical error found in this manuscript.

Author Response

Thank you for your letter and the reviewers’ comments concerning our manuscript entitled "Composition and bioactivity of chlorogenic acids in vegetable and conventional sweet potato vine tips" (ID: foods-2640769). 

We studied the reviewers’ comments carefully and tried to revise the manuscript according to the comments. The following are the responses and revisions we made in response to the reviewers’ questions and suggestions on an item-by-item basis. All the changes are marked in red. Point by point responses to the reviewers’ comments are listed below this letter.

Response Reviewer 3:

Comments 1: As this study aim to investigate the composition and bioactivity of chlorogenic acids in vegetable sweet potato (Shulv No. 1, Fu No. 18, 80 Tainnong No. 71) and conventional sweet potato (Xu Zishu No.8, Xu Shu No.32, Xu Shu 81 No.29). So, the authors should add the photo of vegetable sweet potato compare with conventional sweet potato. In addition, their graphic diversity and habitat should be provided in the manuscript as well.

Response:

Thank you for your comments. The photo information of vegetable sweet potato compare with conventional sweet potato add to the supplement Figure S1.

Comments 2: Line 65-68: As the authors wrote in the manuscript that “In this study, we conducted statistical analysis on vine tip CGAs of different varieties of vegetable sweet potato (Shulv No. 1, Fu No. 18, Tainnong No. 71) and conventional sweet potato (Xu Zishu No.8, Xu Shu No.32, Xu Shu No.29), selected based on their geographic diversity, yield potential and availability for sourcing purposes.”. So, the additional table summarized the information about geographic diversity, yield potential and availability for sourcing purposes of vegetable vs. conventional sweet potato should be provided in the manuscript.

Response:

Thank you for your comments. In the new manuscript we provide additional table S3 summarizing information about geographic diversity, yield potential and availability for sourcing purposes of vegetables and conventional sweet potato.

Comments 3: In scientific name of this plant material should be provide in the introduction of the manuscript.

Response:

Thank you for your comments. The scientific name of the sweet potato has been indicated in the introduction of the new manuscript.

Comments 4: In the figure 6 (B) “Effect of SCGA on migration and invasion ability of breast tumor” should be revised. The higher resolution of figure and the clear bar scale should be added. The current bar scale of figure 6 (B) is very difficult to read.

Response:

Thank you for your comments. The modifications to Figure 6B have been made in accordance with your comments in the revised manuscript.

Comments 5: The greater number of another related published works should be added to discuss with the results from this current study.

Response:

Thank you for your comments. The results and discussion section has been revised in accordance with your comments in the revised manuscript. Relevant published studies were incorporated into the discussion.

Comments 6:There are some spelling mistakes and grammatical error found in this manuscript.

Response:

Thank you for your comments. The manuscript was carefully proofread and corrected for misspellings and grammar.

Reviewer 4 Report

1- A similar study was reported (Analysis of Chlorogenic Acid in Sweet Potato Leaf Extracts) but your article has many experiments which I feel is distracting as it has too much information, therefore, many figures and data should be moved to a supplementary file.

1-     Please provide the photos of all six Samples (also add in the sup. File).

2-     Methodology should be shortened.

3-      Conclusion should be improved

Author Response

Thank you for your letter and the reviewers’ comments concerning our manuscript entitled "Composition and bioactivity of chlorogenic acids in vegetable and conventional sweet potato vine tips" (ID: foods-2640769). 

We studied the reviewers’ comments carefully and tried to revise the manuscript according to the comments. The following are the responses and revisions we made in response to the reviewers’ questions and suggestions on an item-by-item basis. All the changes are marked in red. Point by point responses to the reviewers’ comments are listed below this letter.

Response Reviewer 4:

A similar study was reported (Analysis of Chlorogenic Acid in Sweet Potato Leaf Extracts) but your article has many experiments which I feel is distracting as it has too much information, therefore, many figures and data should be moved to a supplementary file.

Response:

Thank you for your comments. We have transferred some of the tabular data to the supplementary material in the new manuscript.

Comments 1: Please provide the photos of all six Samples (also add in the sup. File).

Response:

We appreciate your valuable comments. The photo information of vegetable sweet potato compare with conventional sweet potato add to the supplement Figure S1.

Comments 2: Methodology should be shortened.

Response:

We appreciate your valuable comments In the new manuscript we have streamlined the material approach to a certain extent.

Comments 3: Conclusion should be improved.

Response:

Thank you for your comments. In the new manuscript, we have revised the conclusion.

Round 2

Reviewer 2 Report

The manuscript is well revised and is now acceptable.